# Checkpoint Inhibitor-Associated Scleroderma and Scleroderma Mimics

**DOI:** 10.3390/ph16020259

**Published:** 2023-02-08

**Authors:** Michael Macklin, Sudeep Yadav, Reem Jan, Pankti Reid

**Affiliations:** 1Section of Rheumatology, Department of Medicine, University of Chicago Medical Center, Chicago, IL 60637, USA; 2Committee on Clinical Pharmacology and Pharmacogenomics, University of Chicago Medical Center, Chicago, IL 60637, USA

**Keywords:** systemic sclerosis, drug toxicities, morphea, fasciitis, autoimmune disease, immune checkpoint inhibitor

## Abstract

Immune checkpoint inhibitors (ICI) are the standard of care for various malignancies and have been associated with a wide spectrum of complications that are phenotypically akin to primary autoimmune diseases. While the literature on these toxicities is growing, there is a paucity of data regarding ICI-associated scleroderma which can carry significant morbidity and limit the ability to continue effective ICI therapy. Our review aimed to analyze the current literature on ICI-associated systemic scleroderma (ICI-SSc) and key scleroderma mimics. Cases of ICI-SSc had notable differences from primary SSc, such as fewer vascular features and less seropositivity (such as scleroderma-specific antibodies and antinuclear antibodies). We found that patients with a diagnosis of SSc prior to the start of ICI can also experience flares of pre-existing disease after ICI treatment used for their cancer. Regarding scleroderma mimics, several cases of ICI-eosinophilic fasciitis have also been described with variable clinical presentations and courses. We found no cases of scleroderma mimics: ICI-scleromyxedema or ICI-scleroedema. There is a critical need for multi-institutional efforts to collaborate on developing a patient database and conducting robust, prospective research on ICI-scleroderma. This will ultimately facilitate more effective clinical evaluations and management for ICI-scleroderma.

## 1. Introduction

Immune checkpoint inhibitors (ICIs) are monoclonal antibodies that have changed the landscape of oncology. They work to decrease the immunosuppressive nature of the tumor microenvironment by blocking immunoregulatory proteins. The most common targets are cytotoxic T lymphocyte–associated antigen 4 (CTLA-4; ipilimumab, tremelimumab), the programmed death 1 receptor (PD-1; nivolumab, pembrolizumab, cemiplimab), and its ligand (PD-L1; atezolizumab, avelumab, and durvalumab). These medications are becoming increasingly common in oncological practice with an ever-expanding list of indications. Nineteen tumor types were included in this list in 2021, in addition to tumor agnostic indications of mismatch repair deficient or high mutation burden tumors [1].

With the robust inhibition of these regulatory steps in the immune system, autoimmunity can be an unintended consequence, causing a wide variety of toxicities which are termed immune-related adverse events (irAEs) [2]. These toxicities are graded by the Common Terminology Criteria for Adverse Events (CTCAE) rubric: grade 1 indicates asymptomatic or mild, grade 2 is moderate, 3 is severe, 4 is life threatening, and grade 5 is an AE resulting in death. The most common irAEs are ICI-hypothyroidism, rash, and diarrhea/colitis [3]. There remains a paucity of data on ICI-induced rheumatic conditions outside of unspecified inflammatory arthritis, polymyalgia rheumatica, and myositis [4]. We aim to review the current literature on one of these areas: ICI-induced systemic sclerosis (ICI-SSc) and scleroderma mimics.

SSc and scleroderma mimics are also occasionally thought to be paraneoplastic in origin and can be induced by several non-ICI cancer chemotherapy agents which provides a potential bias of confounding the ultimate etiology in some cases as well. This significantly limits the conclusions that one can draw from these studies. The nature of the literature on this topic makes it difficult for our intended audience of practicing oncologists, dermatologists, and rheumatologists to gather information from the literature given the lack of large high-quality studies with many small reports. There is no major comprehensive review on the topic in the literature. Our aim is to provide a relevant summary of the literature on ICI-SSc and scleroderma mimics to enable our audience to have an efficient and useful summary of the topic.

The evidence for ICI-SSc and ICI-associated scleroderma mimics is limited to case reports or case series with inherent publication bias and a small number of lower-quality original studies. SSc and scleroderma mimics are already rare entities, rendered less common still when exclusively selecting cases related to ICI use. Articles used in our review were searched for using the database PubMed and Google Scholar. Only English language articles were used, with no exclusion of an article paged on publication date. The following search terms were used: checkpoint and systemic sclerosis, checkpoint and scleroderma, checkpoint and morphea, checkpoint and eosinophilic fasciitis, checkpoint and scleroedema, and checkpoint and scleromyxedema. Reports were reviewed by the study authors and were included if the diagnosis from the reported case seemed probable in our assessment.

In this review, we evaluate how immune checkpoints are involved in the pathogenesis of scleroderma by going through the molecular pathways of immune checkpoints. We also present an overview of scleroderma mimics and survey the available literature for ICI-related cases by summarizing the clinical presentation and treatments attempted in these cases. Using the findings from this review, we provide general recommendations for future evaluations and management of these ICI-related cases of systemic sclerosis and scleroderma mimics.

## 2. Systemic Sclerosis/Scleroderma

The term Scleroderma derives from the Greek roots skleros meaning hard or thickened and derma indicating skin. When organ systems beyond the skin are impacted, the disease is categorized as systemic sclerosis (SSc). SSc is characterized by autoimmune-mediated fibrosis of the skin and internal organs with a variable clinical presentation and severity [5]. SSc remains incompletely understood with a complex pathophysiology [5]. There are three main clinical subsets: (1) diffuse SSc (widespread sclerosis of the skin extending proximal to the elbow/knee), (2) limited SSc (sclerosis is confined to the distal extremities), and (3) SSc sine scleroderma (organ fibrosis without any skin involvement at all) [5].

While the classification is based on the skin findings, the overall clinical course and prognosis also diverge between these groups. Diffuse SSc tends to have a more severe progression of skin fibrosis and higher incidence of interstitial lung disease (ILD). In contrast, the morbidity and mortality of limited SSc are generated from the vascular complications of Raynaud’s and pulmonary arterial hypertension (PAH) [5]. ILD and PAH represent the two major contributions of mortality from SSc, being responsible for 33% and 28% of deaths in SSc patients, respectively [6].

For both diffuse and limited SSc, classic physical examination findings include sclerodactyly (thickening of the fingers/toes) and the prayer sign (inability to fully extend the fingers when placing the palms together) as demonstrated in Figure 1. Aside from sclerodactyly, SSc can lead to significant esophageal dysmotility from fibrotic infiltration, with severe gastroesophageal reflux disease (GERD) symptoms that carry significant morbidity. There are emerging data that GERD influences the development of pulmonary fibrosis through damage from the chronic aspiration of gastric contents with this likely representing at least part of the pathophysiology for SSc-associated ILD [7]. Key CT findings that can help discriminate SSc-related ILD from other causes are fibrosis within ground glass opacities in the upper or lower lobes and reticulations within the lower lobes [8]. SSc associated ILD is an active area of research with unmet need for additional therapeutic agents. Recently the anti-IL-6 monoclonal antibody tocilizumab was approved for slowing progression of SSc associated ILD based on the results of the focuSSed trial [9]. The use of rituximab for SSc associated ILD also continues to evolve with recent data from the RECITAL trial showing non inferiority to more traditionally used cyclophosphamide with less side effects [10]. Notably tocilizumab has also been used for other irAEs including inflammatory arthritis, generally in cases refractory to tumor necrosis factor inhibitors [4,11].

Overall, the exact etiology and pathways leading to SSc are still areas of rich and active research and are not well understood. Fibroblast dysregulation with increased extracellular matrix (ECM) secretion resulting in the fibrosis of tissues/organs is a key feature of SSc and is thought to be due to immune-related mechanisms [11]. Fibroblasts being a key cell type involved in skin thickening in SSc is supported by a correlation of higher circulating fibroblast levels in blood samples from patients with higher modified Rodnan skin scores (mRSS) and dermal thickness when measured by ultrasound [11].

While we focused our review on that of ICI-associated scleroderma, it should be noted that several other oncologic agents such as various chemotherapies have also been associated with scleroderma or morphea-like skin reactions. This includes the taxane drug class, pemetrexed, gemcitabine, and bleomycin [12]. The mechanisms for these reactions are poorly understood, though vesican, a large ECM proteoglycan molecule that serves an immune cell trafficking and activation role, appears to be involved in taxane-induced scleroderma/morphea skin involvement [13,14]. 

For the purposes of this review that focuses on ICI-SSc and its mimics, SSc that is diagnosed prior to start of ICI therapy will be known as primary systemic sclerosis or pre-existing systemic sclerosis and abbreviated as pSSc. On the other hand, SSc that is diagnosed after the start of ICI therapy will be known as immune-checkpoint-associated SSc or ICI-SSc. In this review, we will also discuss cases of ICI-associated morphea as well as scleroderma mimics such as eosinophilic fasciitis, morphea, scleromyxedema, and scleroedema.

### 2.1. Systemic Sclerosis and the Pathophysiologic Role of Checkpoints

PD-1 and PD-L1/L2 are important targets of checkpoint inhibitors. Besides the membrane-bound forms of these molecules on T-cells, antigen-presenting cells, and other cell types, there has also been discussion regarding the soluble forms of PD-1 and PD-L1 or PD-L2 known as sPD-1 and sPD-L1 or sPD-L2, respectively [15]. These soluble forms are thought to interact with the PD-1 and PD-L1/L2 pathway by binding to available membrane-bound PD-L1/L2 and PD-1, respectively [15]. sPD-1 is thought to overall function as a blocker of the pathway leading to increased T-cell activation [16]. The actions of sPD-L1 are more complex and less understood at this time [16]. sPD-L1 is generally thought to function as a stimulator of PD-1, leading to decreased T-cell activation [16]. However, it has also been shown that certain high affinity forms of sPD-L1 can act like a partial agonist, competing with PD-L1 for binding to PD-1 and causing the overall increased activation of T-cells by having less suppressive action on binding to PD-1 [17]. sPD-L2 has also been described though its biological function and is even more poorly understood [18]. Low levels of sPD-L2 have been associated with systemic lupus erythematosus development compared with healthy controls, and low levels have also been associated with platinum resistance in ovarian cancer [19,20].

These PD-1 and PD-L1/L2 pathways may be involved in the pathogenesis of pSSc [15]. Compared with healthy controls, sPD-1 levels were found to be higher in patients with diffuse pSSc and lower in patients with limited pSSc [21]. In contrast, sPD-L1 levels were significantly higher in both diffuse and limited pSSc compared with healthy controls [21]. sPD-1 levels were also correlated with higher modified Rodnan skin scores (MRSS) showing more skin thickening and were associated with the presence of finger contractures with no such association found for sPD-L1 levels [21].

In a different study, sPD-1 and sPD-L2 levels were elevated in patients with pSSc compared with healthy controls, with levels also elevated in patients with diffuse scleroderma relative to those with limited scleroderma [15]. In pSSc, sPD-1 and sPD-L2 also correlated with each other [15]. High levels of both sPD-1 and sPD-L2 levels were associated with a diffuse scleroderma subtype, the development of finger ulcers, pulmonary fibrosis, a higher MRSS, positive Scl-70 antibodies, a decreased vital capacity, and a decreased diffusion capacity for CO_2_ The cytokine IL-10, which overall has immunoregulatory effects and is considered an anti-inflammatory cytokine, was positively correlated with levels of PD-L2 expressing B cells taken from patients with pSSc [15]. Treatment with a fusion protein of PD-L2 bound to the Fc region of an IgG molecule acting as a checkpoint inhibitor (PD-L2-Fc) by binding to available PD-1 in cell culture led to an inhibition in IL-10 production [15]. These B cells with a higher expression of PD-L2 also had lower levels of the cytokines IFNγ with IL-4 and IL-17, all of which are tied to proinflammatory pathways and are involved and even targeted in several autoimmune conditions [15]. Incubation with PD-L2-Fc also led to increases in expression of these inflammatory cytokines [15]. The cell-bound checkpoint proteins do appear to be overall involved in immune regulation outside of their role in the tumor microenvironment with the soluble forms sPD-1 and sPD-L2 acting analogously to endogenous checkpoint inhibitors in opposition to their membrane-bound counterparts. A blockade of PD-1 and PD-L1/PD-L2 by therapeutic checkpoint inhibitors for malignancy could therefore be predicted to influence the development of SSc potentially through increases in proinflammatory cytokines and decreases in regulatory cytokines such as IL-10.

### 2.2. Immune Checkpoint Inhibitor Associated with Systemic Sclerosis

New onset SSc development after checkpoint inhibitor use has rarely been reported in the literature, representing less than 10% of all rheumatic-irAEs (and <1% of all type irAEs) [22,23]. One single-center retrospective study identified only three cases of SSc out of a total of forty-three cases of rheumatic-irAEs [23]. A 2020 report analyzed VigiBase, a drug safety reporting database driven by the World Health Organization [22]. They found thirty-five reports of scleroderma and scleroderma mimic reports associated with checkpoint inhibitor use [22]. Four of these cases were classified as ICI-SSc by the authors with the other cases being classified as a mimic of scleroderma [22]. Our review of these cases yielded three of these four cases in our assessment that had skin changes that were consistent with ICI-SSc. These ICI-SSc cases along with the other reports are summarized in Table 1 [24]. A summary of key clinical features for the various cases are calculated in Appendix A. Of note, melanoma was the most common malignancy in ICI-SSc cases.

The four cases of ICI-SSc presented had atypical features compared with pSSc. While about 90–95% of pSSc patients had a positive antinuclear antibody (ANA), only one of the four cases had a positive ANA [25]. ANA-negative SSc has been described to have a lower burden of vascular disease with less pulmonary arterial hypertension, digital ulcers, and fewer telangiectasias [25,26]. Only one of the four cases of ICI-SSc demonstrated the presence of Raynaud’s; this patient was also ANA negative [27]. Additionally, while 60–80% of patients with pSSc are positive for one of three SSc-associated antibodies (Scl-70, centromere, or RNA polymerase III antibodies) [28], none of the cases of ICI-SSc had either a positive Scl-70, centromere, or RNA polymerase III antibody [25]. In one case, the patient had a low titer ANA with an elevated anti-PM/SCL-75, an antibody associated with overlap cases of myositis/SSc [29,30]. This patient was also reported to have significant muscle weakness with signs of proximal muscle atrophy [29].

**Table 1 pharmaceuticals-16-00259-t001:** Immune-checkpoint-inhibitor-associated scleroderma cases.

Author Journal Year	Age/ Sex	Tumor Type	ICI Used	Time to Develop ICI-SSc	Pertinent Clinical Findings	Labs	Histopathology	Treatment of ICI- SSc	ICI Outcome	Follow Up Time	ICI-SSc Outcome	Tumor Outcome
Grant, BMJ case reports, 2021 [30]	60s/F	Metastatic large cell neuroendocrine lung cancer	Atezolizumab	15 months	Skin thickening of bilateral feet to distal legs and fingers to upper arms, dysphonia, dysphagia, restricted oral aperture opening, dry skin of trunk and thighs without Raynaud’s	ANA 1:40, CRP 1.2 mg/dL, ESR 33 mm/h, aldolase of 12.5 U/L, elevated anti-PM/Scl-75 of 40 units/dL with neg Scl-70, RNA polymerase III, U1 RNP and U3 RNP antibodies	Dermal thickening with mild superficial and deep perivascular lymphoplasmacytic infiltrate	Mycophenolate	Stopped 6 months after skin thickening began (12 months of therapy)	About 13 months from SSc presentation	Improvement	Unknown
Barbosa, Mayo Clinic proceedings, 2017 [28]	66/F	Stage IV metastatic melanoma	Pembrolizumab	39 weeks	Fatigue, joint swelling, muscle weakness with atrophy of deltoids and quadriceps, dry skin, skin thickening of forearms + hands + fingers + thighs + legs + feet + face, lack of Raynaud’s or nail capillary abnormalities, EMG with sensorimotor polyneuropathy primarily axonal without myopathy	Negative Scl-70, ANA, centromere antibodies with normal muscle enzymes and ESR with mildly elevated CRP	Mild dermal fibrosis and sclerosis with trapping of adnexal structures and minimal lymphocytic inflammation	Prednisone with poor response followed by IVIG and mycophenolate	Discontinued 3 weeks after presentation	About 25 weeks	Improvement initially of skin changes followed by worsening fatigue, muscle weakness, and appetite resulting in hospice care	Was in complete remission at last oncology visit but patient passed away on hospice care of unknown cause
Barbosa, Mayo Clinic Proceeding, 2017 [28]	79/M	Stage IV metastatic melanoma	Pembrolizumab	15 weeks	Hand and foot stiffness with skin thickening from fingers to wrists bilaterally and the dorsal surfaces of feet with mildly dilated nailfold capillaries, new onset Raynaud’s, dyspnea with rales in left lung base on exam with patchy ground-glass infiltrates in the lower lung fields on CT diagnosed with ICI-induced pneumonitis	Mildly elevated CRP with negative ANA, centromere, and Scl-70 antibodies	Mild perivascular lymphocytic inflammation and deep dermal sclerosis	Prednisone, hydroxychloroquine	Discontinued on presentation and not rechallenged	About 12 weeks	Improvement	Hepatic metastases, unclear if new after holding ICI, switched to radiotherapy
Cho, The Journal of Dermatology, 2019 [25]	70s/M	Malignant Melanoma	Nivolumab	54 weeks	At 48 weeks patient also developed vitiligo, patient presented with paresthesia and skin tightness in all fingers with difficulty with finger flexion. Ultrasound showed thickened subcutaneous tissues in all fingers	Slightly elevated ESR to 19 mm/h with neg ANA/RNA polymerase III, Scl-70, and centromere antibodies	Edema dermal sclerosis	Prednisone	Nivolumab was continued with no pause of therapy	About 9 months	Improvement	Unknown

Abbreviations: BMJ: British Medical Journal; ICI: immune checkpoint inhibitor; Anti-Scl-70: antibody to the scleroderma 70 kD extractable immunoreactive fragment from topoisomerase antigen; CRP: C-reactive protein; ESR: erythrocyte sedimentation rate; RNP: ribonucleoprotein; ANA: antinuclear antigen; ICI-SSc: immune-checkpoint-inhibitor-associated systemic sclerosis; SSc: systemic sclerosis; Anti PM/SCL-75: antibody to the exosome complex.

Overall, the cases of ICI-SSc described in the literature share some similarities but also key differences when compared to cases of pSSc with a lower frequency of Raynaud’s and less seropositivity (Figure 2). The underlying mechanisms for this divergence are unclear, but likely indicate different pathways of disease propagation when induced by ICI use. Ultimately, the approach to therapy will be based on the severity of presentation along with the risks and benefits of continued ICI use. With the limited data available, a one-size-fits-all approach is not possible to recommend.

### 2.3. Pre-Existing Systemic Sclerosis Requiring Immune-Checkpoint-Inhibitor Treatment

Checkpoint inhibitors have been used to treat malignancy in patients with primary systemic sclerosis, which we refer to as pre-existing SSc (pSSc). Two cases found in the literature are summarized in Table 2 [22]. The patient who developed a scleroderma renal crisis illustrates the occasional difficulty in attributing a cause-and-effect relationship between checkpoint inhibition and complications. In this case, steroid use may have contributed to the development of the scleroderma renal crisis given the known association between steroid use and renal crisis [31]. The concomitant use of several chemotherapies and a VEGF inhibitor further confounds the ability attribute causality, especially as VEGF inhibition independently has been tied to proteinuria, worsening renal function, and thrombotic microangiopathy [32].

A retrospective phase IV safety trial examined a small number of patients with pSSc on checkpoint inhibitors [2]. This comprised seventeen patients with NSCLC being the most common cancer type (n = 13), two with head and neck carcinoma, one with melanoma, and one with colorectal carcinoma [2]. There were roughly equal numbers of patients with limited and diffuse subtypes of scleroderma [2]. The checkpoint inhibitors included nivolumab, pembrolizumab, and durvalumab [2]. The median time of diagnosis of scleroderma was 3.7 years before the diagnosis of malignancy [2]. Of note, scleroderma can be associated with malignancy, mainly in patients with RNA polymerase III antibodies [33]. Specifically, RNA polymerase III antibodies in patients with scleroderma carried an odds ratio of 5.83 of developing malignancy within the next 36 months compared to those patients who did not have this antibody in one cohort [33]. This was limited to the first 36 months after scleroderma diagnosis with no significant effect between months 60 and 120 after scleroderma development [33]. Two patients (12%) in this phase IV safety trial had RNA polymerase III antibodies [2]. Three (18%) had centromere and two (12%) had Scl-70 antibodies [2]. These patients were required to have an inactive/stable disease before receiving ICI [2]. In total, 59% of patients developed an irAE during treatment [2]. These were mostly grade 1 or 2 (64% of irAE) with 6% being grade IV [2]. Sites of irAE included rheumatic (24%), digestive (18%), endocrine (18%), lung (12%), and skin (12%) [2]. In terms of scleroderma, 24% of patients developed a disease flare after checkpoint initiation [2]. Of these flares, 18% were grade III and 6% were grade IV irAEs [2]. Grade III flares occurred after a median of ten months of therapy, and the single grade IV flare occurred after six infusions of pembrolizumab [2]. It is unclear what dosing interval each cycle consisted of for pembrolizumab from the available study data [2]. The single grade IV flare involved a new scleroderma renal crisis; in this case, no steroid use was reported [2]. They required cyclophosphamide for treatment and were RNA polymerase III positive, which is a known risk factor for renal crisis [2].

Though the data are limited, drugs targeting PD-1 rather than CTLA-4 seem to be more associated with the flaring of pSSc. Compared to other irAEs, greater caution must be used with high dose steroids in managing these disease flares given the association with the scleroderma renal crisis [31]. Hence, our recommendation is to involve a rheumatologist to devise an individualized treatment plan for each case.

## 3. Localized Scleroderma: Morphea

Morphea or localized scleroderma presents as circumscribed areas of skin thickening without any visceral involvement and spares the distal extremities, which marks this as a distinct entity from systemic sclerosis [34]. Plaque morphea is the most common subtype, involving a well-defined area of thickened skin with a predilection for the trunk or back [34]. Other types include bullous morphea involving superimposed blisters or bullae, deep morphea which involves fat or fascia, generalized morphea which involves plaques on more than two body sites, or linear which involves linear streaks of cutaneous involvement [34]. The etiology of morphea is poorly understood but is thought to be at least partially inflammatory in origin [34]. Treatment generally consists of topical steroids of calcineurin inhibitors, phototherapy, and rarely systemic immunosuppression [34].

Key clinical features of these morphea cases are presented in Table 3 with a numerical summary of these cases reflected in Appendix A [35,36,37,38,39]. From these cases, it is notable that patients were often exposed to the CTLA-4 antagonist ipilimumab as part of their clinical course; however, temporally the morphea did not occur until patients were off ipilimumab and on pembrolizumab. There were also two cases of morphea in patients only exposed to pembrolizumab. Based on the data, we can surmise that PD-1-acting ICIs may be higher risk for this complication than ipilimumab.

Metastatic melanoma was the common tumor type likely owing to ICI use being so prevalent for this malignancy. Despite the limited literature, most cases of ICI-induced morphea can likely be treated with topical therapy initially. The case demonstrated by Herrscher et al. demonstrates a very unusual treatment course, necessitating multiple trials of immunosuppression [37]. This would be atypical for de novo morphea and likely represents an outlier of a particularly severe case. There are limited data to advise whether ICI must be held with the development of morphea, but we suspect in most cases that ICI therapy can be continued with topical therapy given for morphea. More complex cases would benefit from the involvement of dermatology and/or rheumatology if there were suspicion that morphea may not be the correct diagnosis or if systemic immunosuppression is required.

## 4. Scleroderma Mimics

### 4.1. Eosinophilic Fasciitis

Eosinophilic fasciitis is a rare fibrosing disorder consisting of erythema, edema, and the induration of the tissues of the extremities, and it is considered a scleroderma mimic [40]. Classic findings include the peau d’orange appearance of the involved skin and a “groove sign” consisting of a linear depression along the course of the superficial veins due to tethering [40,41]. Patients often have peripheral eosinophilia and this disease has been associated with monoclonal gammopathy and solid malignancies [40]. The exact etiology is unknown [40]. No widely accepted criteria exist, but proposed diagnostic criteria include the Pinal-Fernandez criteria and the Jinnin criteria [41,42]. Notably, the Jinnin criteria highlights the absence of Raynaud’s phenomenon to separate this disorder from scleroderma [42].

Eosinophilic fasciitis can be induced by checkpoint inhibitors with many cases found in the literature. All cases are summarized in Table 4, with a statistical summary of the key clinical features in Appendix A [43,44,45,46,47,48,49,50,51,52,53]. Based on the limited data available, checkpoint-inhibitor-induced eosinophilic fasciitis appears to be more tied to PD-1/PD-L1 inhibition and not CTLA-4 inhibition. Data for eosinophilic fasciitis treatment independent of checkpoint inhibition are scarce and are mainly based on case series/reports and some retrospective studies [40]. From the data available, our recommendation would be to hold ICI and use steroids as first-line therapy with methotrexate as an add on for resistant cases mirroring the standard treatment of de novo eosinophilic fasciitis [40]. There is not enough information as to whether these patients can be rechallenged with checkpoint inhibitors once the improvement of eosinophilic fasciitis occurs, and we would suggest this be individualized based on patient goals and values after a discussion of potential benefits and risks with a patient and their oncologist.

### 4.2. Scleroderma and Scleromyxedema

Scleromyxedema is a rare fibro mucinous disease in the differential when scleroderma is suspected [54,55]. It is unique in that it is strongly associated with monoclonal gammopathy [54,55]. It follows a similar distribution of skin involvement of the hands and face as scleroderma, though with the distinctive appearance of firm translucent papules instead of smooth thickening [54,55]. Histology shows the presence of dermal mucin deposition and increased collagen with fibroblast proliferation [54,55]. Additionally unique to scleromyxedema is the nervous system’s involvement, which occurs in about 10–30% of diagnoses [54,55]. A severe presentation known as dermatoneuro syndrome which manifests as a sudden onset altered consciousness, gait issues, fever, evolving to seizures as sometimes death have been described [54,55]. Likely owing to it being related to a paraneoplastic origin, no cases have been attributed to ICI use in the literature.

Scleroedema is another fibrosing cutaneous disorder associated with thickening around the neck spreading to the upper parts of the body without the involvement of the limbs (in contrast to scleroderma) [56]. Raynaud’s and autoantibody positivity are also not seen in scleroedema [56]. Three types have been described, including a self-resolving postinfectious form, a progressive form associated with infections or monoclonal protein and a slowly progressive type that occurs in diabetic patients likely due to the glycosylation of collagen [56]. Histology is significant for a greatly thickened dermis with normal epidermis [56]. Scleroedema has not been associated with ICI use in the literature, likely owing to its non-autoimmune-mediated mechanisms of disease.

While these two disease states have not been associated with ICI use, they are important to keep in the differential in a patient who develops skin lesions that are suspicious for scleroderma. Patients may have monoclonal gammopathies in addition to their malignancy being treated with ICI or have concomitant diabetes mellitus that makes them at risk for developing one of these scleroderma mimics. If it is truly a paraneoplastic disorder rather than a drug-induced phenomenon, it is essential to continue effective cancer therapy. A multidisciplinary collaboration for clinical work up and management for these complex cases is essential for optimal patient care.

## 5. Conclusions

The breadth of irAEs-mediated disease states continues to expand as ICI use becomes more customary throughout the tumor spectrum. SSc and scleroderma mimics represent a group of disorders with significant morbidity that have been described with wide clinical presentations because of ICI use, with ICI use also being described as resulting in the flaring of pSSc. Oncologists, dermatologists, and rheumatologists should be aware of these potential ICI toxicities. Compared with pSSc, ICI-SSc-induced cases appear to have fewer vascular features with a reduced frequency of Raynaud’s, as well as lower levels of scleroderma-specific antibody and ANA positivity. Cases to date suggest PD-1/PD-L1 blockades to be higher risk than CTLA-4 inhibition. ICI use has also have been described to cause flares of pSSc. Scleroderma mimics such as morphea and eosinophilic fasciitis have been reported with a similarly wide range of presentations, treatments, and clinical outcomes. No ICI-related cases of scleromyxedema and scleroedema were found in our literature review. However, due to their association with malignancy and other common comorbidities, these entities should be included on the differential in a patient that develops skin thickening during ICI use to avoid unneeded and potentially harmful changes to their cancer treatment regimens.

Further studies and reports of cases will help us understand whether the clinical finding that most ICI-SSc cases have a lower burden of vascular features and autoantibody positivity compared with primary SSc cases remains consistent. Our review serves as a starting point on this topic and represents a complete and comprehensive summary of the field’s current knowledge and description of ICI-related SSc and scleroderma mimics. Our hope is that this review will be a useful reference for currently practicing physicians who treat ICI-related irAEs, representing a practical source of information compared with perusing the of dozens of individual case reports/series in the literature. Such a source would also be useful in spurring further research on this topic. Treatment guidance at this time is limited, ranging from the continuation of ICI to drug cessation and the addition of immunosuppression. We recommend extrapolating from the standardized management of pSSc and related conditions, with the decision to hold or resume ICI decided on a case-by-case basis weighing the severity of the reaction against the options and outcomes expected for the underlying cancer. As more cases are described, we would look forward to consensus guidelines on the management of ICI-induced scleroderma, morphea, and eosinophilic fasciitis. Multicenter international clinical trials for the treatment of these complications would be ideal to optimize treatment strategies and help forge this consensus. As the use of these agents broadens and expands, this represents an exciting avenue of future research for the field of ICI-induced autoimmunity.

## Figures and Tables

**Figure 1 pharmaceuticals-16-00259-f001:**
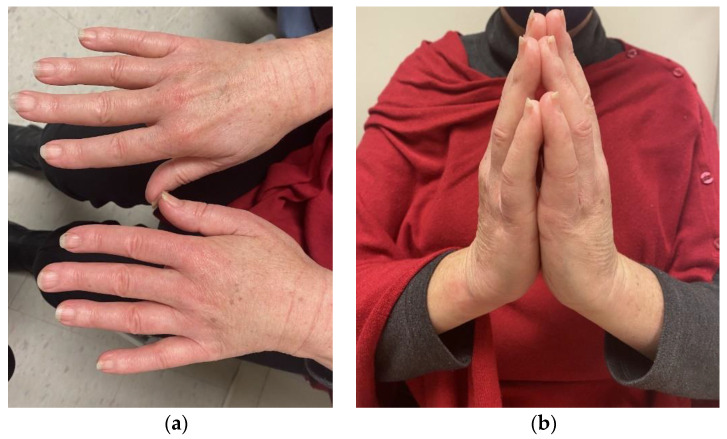
Physical exam findings associated with systemic sclerosis. Pertinent findings associated with systemic sclerosis include notable skin tightening with sclerodactyly (**a**) and consequent limitations in “prayer sign” (**b**) that is demonstrated here by a patient who received diagnosis of systemic sclerosis after start of checkpoint inhibitor therapy.

**Figure 2 pharmaceuticals-16-00259-f002:**
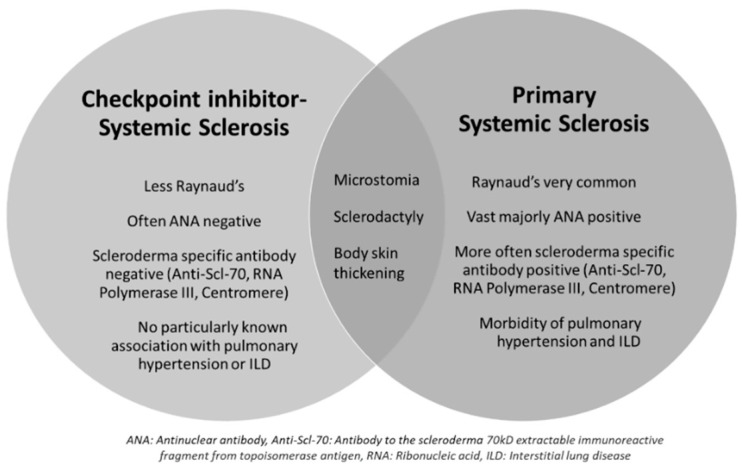
Similarities and differences between ICI-SSc and primary SSc. This Venn diagram highlights key differences and similarities between immune-checkpoint-inhibitor-associated systemic sclerosis (ICI-SSc) and primary systemic sclerosis (SSc). Abbreviations—ANA: antinuclear antibody; Anti-Scl-70: antibody to the scleroderma 70 kD extractable immunoreactive fragment from topoisomerase antigen; RNA: ribonucleic acid; ILD: interstitial lung disease.

**Table 2 pharmaceuticals-16-00259-t002:** Immune checkpoint inhibitor use in preexisting scleroderma cases.

Author Journal Year	Age/ Sex	Tumor Type	ICI Used	Duration of ICI That Led to Worsening of PSSc	Pertinent Clinical Findings	Pertinent Labs	Histopathology	Treatment of PSSC	ICI Outcome	Follow Up Duration	PSSc Outcome	Tumor Outcome
Terrier, Autoimmunity Reviews, 2020 [23]	49/W	Non-small-cell lung cancer	Pembrolizumab	3 weeks	Pre-existing PSSc based on symptoms of GERD, Raynaud’s, puffy fingers before ICI use. Developed dramatic increase in thickening of trunk, abdomen, and limbs. Mild to moderate nailfold abnormalities seen. Prednisone trialed with continued thickening of the breasts and back	ANA 1:640, negative Scl-70, centromere, and RNA Poly III	Swollen collagen bundles in reticular dermis and a minimal inflammatory infiltrate	Prednisone (failed) followed by prednisone and cyclophosphamide	Held 3 weeks after skin thickening began	About 9 weeks	Unknown	Initial tumor shrinkage on CT but unknown after
Terrier, Autoimmunity Reviews, 2020 [23]	57/W	Non-small-cell lung cancer	Pembrolizumab with exposure to bevacizumab, pemetrexed, cisplatin, docetaxel	18 weeks	NSLC was diagnosed during CT to screen for ILD for limited scleroderma. Patient was treated with 10 mg of prednisone at diagnosis for arthralgia. Patient developed diffuse extension of skin thickening to proximal limbs and trunk and developed scleroderma renal crisis	Positive ANA of unknown titer, positive RNA poly III	None	Prednisone followed by cyclophosphamide, angiotensin-converting enzyme inhibitors	Stopped when worsening skin thickening began without rechallenge	About 24 weeks from onset of worsening PSSc	Improvement	Unknown

Abbreviations: ICI: immune checkpoint inhibitor; Anti-Scl-70: antibody to the scleroderma 70 kD extractable immunoreactive fragment from topoisomerase antigen; ANA: antinuclear antigen; ICI-SSc: immune-checkpoint-inhibitor-associated systemic sclerosis; PSSc: primary systemic sclerosis; ILD: interstitial lung disease; CT: Computerized Tomography; NSLC: non-small-cell lung cancer; GERD: gastrointestinal esophageal reflux disease.

**Table 3 pharmaceuticals-16-00259-t003:** Description of immune-checkpoint-inhibitor-induced morphea cases.

Author Journal Year	Age Sex	Tumor Type	ICI Used	Duration of ICI That Led to Worsening of Preexisting Morphea	Pertinent Clinical Finding/ Other Complications	Pertinent Labs	Histopathology	Treatment of Morphea	ICI Outcome	Follow Up Duration	Morphea Outcome	Tumor Outcome
Acar, Journal of cosmetic dermatology, 2020 [36]	48/F	Metastatic melanoma	Ipilimumab with progression followed by nivolumab	66 weeks of nivolumab	Patient also developed hypothyroidism and vitiligo before morphea. Developed a whitish-colored sclerotic plaque on the left supraclavicular area	ANA of 1:320 with neg Scl-70	Epidermal atrophy, coarsening and homogenization of dermal collagen, and interstitial lymphoid cell infiltration	Mometasone and calcipotriol	Continued	9 months	Improvement	Stable disease
Montero-Menárguez, Dermatitis, 2022 [37]	53/F	Rectal adenocarcinoma	2 cycles of nivolumab and ipilimumab followed by 6 months of nivolumab monotherapy	7 months after first dose of nivolumab	Pruritic erythematous plaques which evolved after 1 week to oval and indurated yellowish–brownish shiny plaques distributed predominantly on the trunk and proximal extremities	Neg for ANA and unknown tested immuoserology	Preserved epidermis, sclerotic fibrosis, and a slight perivascular lymphocytic infiltrate of the superficial and middle reticular dermis	Clobetasol and prednisone 30 mg followed by worsening skin lesions on 20 mg followed by ICI discontinuation and initiation of hydroxychloroquine	Stopped after failure to wean steroids without restarting	Unknown	Improvement	Complete response
Zafar, Melanoma research, 2021 [40]	31/F	Metastatic melanoma	Pembrolizumab with IDO inhibitor followed by experimental TLR9 activator injections followed by pembrolizumab	1 year after final pembrolizumab monotherapy dose or 3 years after pembrolizumab/IDO combination therapy initiation	Patient had a complete response to ICI with no active tumor before morphea. White lesion on right ankle initially followed by numerous oval erythematous plaques on the trunk, some of which had yellow–white waxy centers	None described	None described, though punch biopsy performed “consistent with morphea”	None described	Patient had already finished ICI therapy	3 months from initial morphea presentation	None described	Had complete response before morphea
Herrscher, European journal of cancer, 2019 [38]	74/F	Metastatic melanoma	Pembrolizumab	30 weeks	Developed vitiligo, arthralgias, and pruritis/burning skin at 21 weeks post-ICI. Presented with painful and atrophic skin on neck, chest, shoulders	Unknown but reported negative immuoserology	Thickened collagen fibers in the dermis with a dermal lymphocytic infiltration	Clobetasol/colchicine followed by prednisone followed by cyclophosphamide followed by rituximab	Discontinued at presentation without restarting	7 months from initial morphea presentation	Improvement after several lines of therapy	Continued complete response
Cheng, International journal of dermatology, 2019 [39]	64/M	Metastatic melanoma	Ipilimumab followed by pembrolizumab	15 weeks of pembrolizumab	Developed hypothyroidism and vitiligo and ICI-induced colitis on Ipilimumab previously. Presented with violaceous hyperpigmented, shiny plaques involved the upper chest, back, left abdomen, bilateral arms, and bilateral shins with sparing of the hands, feet, and face. May have had skin thickening of left upper arm 8 years prior which spontaneously resolved	ESR of 38 mm/h, no eosinophilia, neg ANA, centromere, and Scl-70 antibodies	Increased sclerotic collagen bundles in the dermis and subcutis and a CD8 predominant lymphocytic infiltrate with plasma cells and eosinophils	Prednisone	Pembrolizumab stopped and not restarted	Unclear duration	Improvement	New lung nodules followed by spontaneous resolution of nodules on PET

Abbreviations: ICI: immune checkpoint inhibitor; Anti-Scl-70: antibody to the scleroderma 70 kD extractable immunoreactive fragment from topoisomerase antigen; ANA: antinuclear antigen; ESR: erythrocyte sedimentation rate; IDO: Indoleamine 2,3-dioxygenase; TLR9: Toll-like receptor 9.

**Table 4 pharmaceuticals-16-00259-t004:** Cases of eosinophilic fasciitis in the setting of immune checkpoint inhibitors use.

Author Journal Year	Age Sex	Tumor Type	ICI Used	Duration of ICI That Led to Eosinophilic Fasciitis	Pertinent Clinical Finding/Other Complications	Pertinent Labs	Histopathology	Treatment of Eosinophilic Fasciitis	ICI Outcome	Follow Up	Eosinophilic FASCIITIS Outcome	Tumor Outcome
Salamaliki, Rheumatology and therapy, 2020 [44]	81/M	Non-small-cell lung	Pembrolizumab	13 months	Patient was diagnosed with “scleroderma” by another physician, which authors disagreed with. Presented with sclerotic skin lesions of his legs and forearms sparing his fingers and a linear depression along the course of his veins on his forearms consistent with a groove sign with lack of Raynaud’s, arthralgia	ANA 1:160 with normal eosinophil levels	Thickening and hyalinization of the connective tissue of the deep dermis, subcutaneous fat and muscular fascia, perivascular and focal interstitial lymphocytic and plasma cell infiltrates in the subcutaneous fat and localized atrophy of the adnexal structures, without eosinophilic infiltrates	16 mg of methylprednisolone and mycophenolate for “scleroderma” followed by methylprednisolone monotherapy with taper	Discontinued followed by restarting 2.5 months later	Unclear	Improved	Complete response
Parker, BMJ case reports, 2018 [45]	43/W	Metastatic melanoma	Nivolumab	14 months	Developed autoimmune thyroiditis 6 months after ICI initiation. Presented with widespread myalgia of all limbs with fatigue, progressive dysphagia, borderline EMG, and diminished proximal muscle strength. Skin over forearms, calves, and chest had “woody” feeling. MRI with symmetric fascial thickening of all thigh and calf muscle groups	Normal CK, normal eosinophil count, neg screen for myositis-specific and -overlap connective tissue antibodies	Initial unremarkable percutaneous tibialis anterior muscle biopsy. Repeat full thickness skin–muscle biopsy with fascial and perifascicular inflammatory infiltrate with upregulation of MHC I on myofibers	Prednisolone followed by IVIG	Unclear if held or not	At least months, unclear total	Mild improvement	Complete response
Chan, The Oncologist, 2020 [46]	48/M	Stage IV lung adenocarcinoma	Atezolizumab with erlotinib	6 months	Tightness and pain in his upper and lower extremities accompanied by leg swelling followed by thickening of the forearms and legs distal to the knees with a groove sign present on the left leg MRI of left tibia showed mild fascial edema	Initially CK of 933 U/L followed by spontaneous resolution, Eosinophil count of 700–3500	Expansion of the fascia by collagen, hyaluronic acid, and fibrin accompanied by numerous lymphocytes and plasma cells without tissue eosinophilia	Prednisone and methotrexate with discontinuation of methotrexate 1 month later due to chemo interaction	Three months after skin symptoms began it was discontinued and not restarted	Three years	Improved	Progression with response to other therapy
Chan, The Oncologist, 2020 [46]	71/F	Vulvar Melanoma	Nivolumab	3 months	Myalgias involving the shoulders, thighs, and calves, pitting edema in the arms and feet followed by 1 month later with skin thickening in a circumscribed area on each forearm just proximal to the wrists followed by extension of thickening to her forearms circumferentially with a woody texture with tethering seen	Eosinophil count peaked at 2400	Poikiloderma on initial punch biopsy. Full thickness biopsy with inflammatory fibrosing reaction involving the subcutaneous tissue, fascia, and skeletal muscle with inflammatory cell infiltrate consisted of lymphocytes, plasma cells, and eosinophils	Prednisone and infliximab followed by prednisone and methotrexate followed by prednisone monotherapy	Held at beginning of symptom onset without restarting	Twelve months	Halted progression without improvement	Complete response
Chan, The Oncologist, 2020 [46]	43/M	Metastatic melanoma	Pembrolizumab	15 months	Presented with subjective tightness and swelling of the forearms. Later, noted to have limited wrist mobility bilaterally and swelling of the flexor aspect of the forearms with normal appearance of the skin MRI of the right forearm showed mild tenosynovitis in the flexor and extensor compartments, with fascial edema	His CK was normal, and eosinophil count was 700	No biopsy performed	Prednisone and mycophenolate	Held 4 weeks after presentation without restarting	3 months	Improvement	Unknown
Wissam, Journal of immunotherapy and precision oncology, 2019 [47]	48/F	Triple negative breast cancer	nab-paclitaxel and atezolizumab	60 weeks	Edema of lower extremities followed by skin pain, worsening edema, and erythema with thickening of the skin sparing the fingers and toes. MRI of the distal extremities showed hyperintense signal of the subcutaneous adipose tissue suggestive of fasciitis	Eosinophil count of 1400 with normal CK, normal CRP/ESR, neg CCP and RF	Thickened fibrotic fascia with signs of fasciitis, chronic inflammation, and panniculitis	None besides ICI cessation	Atezolizumab was stopped 20 weeks after symptom onset without restarting	72 weeks	Improvement	Durable tumor response
Khoja, Cancer immunology miniatures, 2016 [48]	51/F	Metastatic melanoma	Pembrolizumab	19 months	Symptoms began one month after finishing ICI with complete response of tumor. Patient complained of muscle aches and heaviness in the limbs with symptom progression over the next 6 weeks with new headaches accompanied by floaters in her visual fields. Six weeks later had visible puffiness of the face and thickened and tethered waxy skin on all limbs and on the abdomen. An MRI of the right upper limb revealed marked fascial edema associated with the musculature of the arm and the right chest wall involving the latissimus dorsi, serratus anterior, and pectoralis muscles	Peak eosinophil count of 5240 with normal ESR, CK	A full-thickness biopsy of skin and subcutaneous tissue showed infiltration of the dermis with a lymphoeosinophilic infiltrate with scattered eosinophils in the interstitium with the fascia containing a denser infiltrate of eosinophils, plasma cells, and lymphocytes	Methylprednisolone and prednisone	Already completed ICI therapy one month prior to symptom start	14 weeks	Improvement	Unknown
Lidar, Autoimmunity reviews, 2018 [49]	53/F	Metastatic melanoma	Pembrolizumab	8 months	Unknown clinical features	Unknown	A full-thickness biopsy of skin and subcutaneous tissue showed infiltration of the dermis with a lymphoeosinophilic infiltrate with scattered eosinophils in the interstitium with the fascia containing a denser infiltrate of eosinophils, plasma cells, and lymphocytes	Methylprednisolone and prednisone	Held and not restarted	6 months	Improved	Complete response
Le Tallec, Journal of thoracic oncology, 2019 [51]	56/M	Stage IV pulmonary adenocarcinoma	Nivolumab	9 months	Presented with myalgia and a diffuse skin thickening. Prednisone was started, and once he reached 20 mg of a taper his myalgia and eosinophilia, he returned with MRI showing abnormal signal along the fascia. He may have also developed immune mediated cholangitis	Eosinophil count was 4140 with normal CK, ANA of 1:320	Marked CD8-positive inflammatory infiltrate of the fascia coexisting with eosinophils	Prednisone and methotrexate	Held at symptom onset without restarting	5 months	Improvement	Complete response
Ollier, Rheumatology advances in practice, 2020 [52]	64/M	Metastatic melanoma	Nivolumab	52 weeks	Initially had progressive fatigue and proximal weakness with edema of the lower limbs followed by the presence of a groove sign on his upper limb and arthralgia. EMG conducted was normal. MRI showed thickening and enhancement of the fascia in the medial and posterior muscle compartments of the lower limbs	Eosinophil count of 1800 and CRP of 115 with normal CK	Thickened fibrotic fascia with signs of fasciitis, chronic inflammation, and panniculitis	Prednisone followed by prednisone/methotrexate followed by the addition of IVIG	Discontinued 5 months after symptom onset without restarting	About 17 months	Improvement	Unknown but response noted part way through follow up time
Andres-Lencina, Australasian college of dermatologists, 2018 [53]	65/M	Stage IV bladder cancer	Nivolumab/ipilimumab for 12 weeks followed by nivolumab monotherapy	56 weeks after nivolumab monotherapy or 68 weeks after combination therapy	Developed lichen sclerosis of glans penis 4 weeks after nivolumab monotherapy or 16 weeks after combination therapy. Then developed an asymptomatic brownish red plaque with significant induration on the pubis that extended to the left anterior iliac crest region	Eosinophil count of 4000	A full-thickness biopsy of skin and subcutaneous tissue was performed, which showed dermal fibrosis with dense hyalinized collagen with lymphocytic perivascular infiltration with eosinophils that went to the fascia	Steroids followed by cyclosporine due to GI bleed on steroids followed by methotrexate	Held and not restarted	5 months	Improvement on prednisone then without effective response to ciclosporin and methotrexate	Progression and death from cancer
Le Tallec, Journal of thoracic oncology, 2019 [51]	56/M	Stage IV pulmonary adenocarcinoma	Nivolumab	9 months	Presented with myalgia and a diffuse skin thickening. Prednisone was started and once he reached 20 mg of a taper his myalgia and eosinophilia returned with MRI showing abnormal signal along the fascia. He may have also developed immune-mediated cholangitis	Eosinophil count was 4140 with normal CK, ANA of 1:320	Marked CD8-positive inflammatory infiltrate of the fascia coexisting with eosinophils	Prednisone and methotrexate	Held at symptom onset without restarting	5 months	Improvement	Complete response
Ollier, Rheumatology advances in practice, 2020 [52]	64/M	Metastatic melanoma	Nivolumab	52 weeks	Initially had progressive fatigue and proximal weakness with edema of the lower limbs followed by the presence of a groove sign on his upper limb and arthralgia. EMG conducted was normal. MRI showed thickening and enhancement of the fascia in the medial and posterior muscle compartments of the lower limbs	Eosinophil count of 1800 and CRP of 115 with normal CK	Thickened fibrotic fascia with signs of fasciitis, chronic inflammation, and panniculitis	Prednisone followed by prednisone/methotrexate followed by the addition of IVIG	Discontinued 5 months after symptom onset without restarting	About 17 months	Improvement	Unknown but response noted part way through follow up time
Andres-Lencina, Australasian college of dermatologists, 2018 [53]	65/M	Stage IV bladder cancer	Nivolumab/ipilimumab for 12 weeks followed by nivolumab monotherapy	56 weeks after nivolumab monotherapy or 68 weeks after combination therapy	Developed lichen sclerosis of glans penis 4 weeks after nivolumab monotherapy or 16 weeks after combination therapy. Then developed an asymptomatic brownish red plaque with significant induration on the pubis that extended to the left anterior iliac crest region	Eosinophil count of 4000	A full-thickness biopsy of skin and subcutaneous tissue was performed, which showed dermal fibrosis with dense hyalinized collagen with lymphocytic perivascular infiltration with eosinophils that went to the fascia	Steroids followed by cyclosporine due to GI bleed on steroids followed by methotrexate	Held and not restarted	5 months	Improvement on prednisone then without effective response to ciclosporin and methotrexate	Progression and death from cancer

Abbreviations: BMJ: British Medical Journal; CK: Creatine kinase; EMG: electromyography; MRI: magnetic resonance imaging; ICI: immune checkpoint inhibitor; ANA: antinuclear antigen; IVIG: intravenous immunoglobulin; CRP: C-reactive protein; ESR: erythrocyte sedimentation rate; MRI: magnetic resonance imaging; RF: rheumatoid factor; CCP: antibody against Cyclic Citrullinated Peptide; PET: positron emission tomography.

## Data Availability

Data sharing not applicable.

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
