# Peer review of "Checkpoint Inhibitor-Associated Scleroderma and Scleroderma Mimics"

_pharmaceuticals, 2023, doi:10.3390/ph16020259_

Round 1

Reviewer 1 Report

This manuscript presents a systematic review related with checkpoint inhibitor associated scleroderma and scleroderma  mimics. This review  is generally well written, but it is unclear what this review adds to what is already known and have been published earlier. No clear research question seems to be formulated, the conclusions are unclear and other major concerns with this manuscript.

My specific comments are stated below. Overall, several important issues need to be addressed and some are of methodological character which requires a considerable revision of the paper. 

1. The introduction section did not provide a clear rationale for carrying out the study (for example, why is your research question important? What gap in the literature is the study addressing?

2. Methods: Was the study registrered at PROSPERO? Please to include the date of registered .

3.  Methods - literature search and selection: Please outline the exact search string or provide an appendix with the search strategy with specific search outcomes for each search and combinations. 

4. Methods - literature search and selection: Did you restrict study selection on any language? 

5. Results.The authors have not performed a systematic review, according to international standards, so they do not provide specific numerical data. Please add the calculate related with risk of bias was evaluated of this investigation.

6. Within your discussion,  compare outline your results, discuss their novelty and their application to practice.

 7. Conclusion. These conclusions need to be softened, modified a in order to reflect only the study findings.

Author Response

Thank you so much for taking the time to read our manuscript and for your valuable feedback. We sincerely appreciate the opportunity to re-submit for your consideration our manuscript (2170743), entitled “Checkpoint inhibitor associated scleroderma and scleroderma mimics for re-consideration for publication in Pharmaceuticals.

Please see our responses under your reviewer suggestions and feedback below:

REVIEWER: 
This manuscript presents a systematic review related with checkpoint inhibitor associated scleroderma and scleroderma  mimics. This review  is generally well written, but it is unclear what this review adds to what is already known and have been published earlier. No clear research question seems to be formulated, the conclusions are unclear and other major concerns with this manuscript.

My specific comments are stated below. Overall, several important issues need to be addressed and some are of methodological character which requires a considerable revision of the paper. 

REVIEWER COMMENT 1:
The introduction section did not provide a clear rationale for carrying out the study (for example, why is your research question important? What gap in the literature is the study addressing?

Authors:

The introduction has been expanded/modified. In a nutshell our goal/intent is to have a comprehensive review on the subject providing a useful reference for future research and currently practicing physicians. Currently in the literature no large reviews exist on this topic. Our review represents a comprehensive summary of useful information for these purposes and does attempt to gather general trends/differences in these cases such as decreased vascular features/antibody positivity in ICI induced SSc cases described in the paper. 

REVIEWER COMMENT 2:
Methods: Was the study registered at PROSPERO? Please include the date of registered .

Authors:

We did not attempt to perform a systemic literature review and performed a general review regarding immune checkpoint inhibitor associated systemic sclerosis and scleroderma mimics. Because this was not meant to be a systematic literature review, it was not registered in PROSPERO.

REVIEWER COMMENT 3:
Methods - literature search and selection: Please outline the exact search string or provide an appendix with the search strategy with specific search outcomes for each search and combinations. 

Authors:

Thank you for the recommendation. As we did not conduct a systematic literature review, we did not dedicate a whole section to methodology. To relay our search strategy to our readership, we added information in our manuscript regarding the search terms used for our review:

“Articles used in our review were searched for using the database PubMed and Google Scholar. Only English language articles were used, with no exclusion of an article based on publication date. The following search terms were used: checkpoint and systemic sclerosis, checkpoint and scleroderma, checkpoint and morphea, checkpoint and eosinophilic fasciitis, checkpoint and scleredema, checkpoint and scleromyxedema. Reports were reviewed by the study authors and were included if the diagnosis from the reported case seemed probable in our assessment.”

REVIEWER COMMENT 4:
Methods - literature search and selection: Did you restrict study selection on any language? 

Authors:

We only used papers/reports that were available in English. We clarified this in our manuscript.

REVIEWER COMMENT 5:
The authors have not performed a systematic review, according to international standards, so they do not provide specific numerical data. Please add the calculate related with risk of bias was evaluated of this investigation.

Authors:

Thank you for your comment. We did not set out to do a systematic literature review, but a general review. This is often done in the field of medicine. A systemic review would be challenging given the low-quality evidence available (which is mostly case reports or series) with a high chance of confounding that we openly acknowledge in the paper. We do not have formal risk of bias calculations given this, but we do reflect in our manuscript that the data quality is low.

REVIEWER COMMENT 6:
Within your discussion,  compare outline your results, discuss their novelty and their application to practice.

Authors:

We appreciate this suggestion. This has been expanded upon. In general, we do not intend this to be the final word on the topic. This is meant to be a useful source for future research or practicing physicians to be able to analyze what the current data is on ICI induced SSc and mimics. Please see additional part of discussion below which only includes newly added material. Thank you.

“Further study and reporting of cases will help us understand whether the clinical finding that most ICI-SSc has a lower burden of vascular features and autoantibody positivity compared with primary SSc cases remains consistent. Our review serves as a starting point on this topic and represents a complete and comprehensive summary of the field’s current knowledge and description of ICI related SSc and scleroderma mimics. Our hope is that this review will be a useful reference for currently practicing physicians who treat ICI-related irAEs, representing a practical source of information compared with perusing the dozens of individual case reports/series in the literature. Such a source would also be useful in spurring further research on this topic.”

REVIEWER COMMENT 7:
These conclusions need to be softened, modified in order to reflect only the study findings.

Authors:

We appreciate this recommendation and made sure that our conclusions are fairly soft. Our main purpose is to provide a useful review for our audience. Conclusions from our literature review, such as that ICI induced and primary SSc have key clinical differences including less antibody positivity and less vascular features is supported in our paper and from our references. Our treatment recommendations are fairly conservative and based on expert opinion, with three of the authors being practicing rheumatologists.

Reviewer 2 Report

I thank the academic editor for giving me the opportunity to review this very interesting manuscript in which the authors conduct a careful analysis regarding the conditions of scleroderma and scleroderma mimics in patients treated with immune checkpoint inhibitors (ICIs). I think the paper is well written, clear and very comprehensive (including the attached material), and I believe it can be published after further checking of the English language and a check of some typos present.

Author Response

Dear Reviewer, 

Thank you so much for taking the time to read our manuscript and for your comments. We sincerely appreciate the opportunity to re-submit for your consideration our manuscript (2170743), entitled “Checkpoint inhibitor associated scleroderma and scleroderma mimics for publication in Pharmaceuticals.

Sincerely, 

Michael Macklin, Sudeep Yadav, Reem Jan and Pankti Reid

Reviewer 3 Report

1) There is a critical need for multi-institutional ef- forts to collaborate on developing a patient database for and conducting robust, prospective re-  search on ICI-scleroderma. This will ultimately facilitate more effective clinical evaluation and man- agement for ICI-scleroderma. Please underline the results of the review.

2) Introduction. L52-56. SSc and scleroderma  mimics are already rare entities, rendered less common still when exclusively selecting cases related to ICI use. Pleae, add a brief paragraph on systemic sclerosis (SSc) and scleroderma  mimics and insert this references:

a-Correlation between circulating fibrocytes and dermal thickness in limited cutaneous systemic sclerosis patients: a pilot study. Rheumatol Int. 2019;39(8):1369-1376. doi:10.1007/s00296-019-04315-7

b- Hyperpigmented Scleroderma-like Lesions under Combined Pembrolizumab and Pemetrexed Treatment of Non-Small Lung Cancer. Dermato 20222, 8-13. https://doi.org/10.3390/dermato2010002

3) In this review, we review the specific role of immune checkpoints in the  pathogenesis of scleroderma. We further summarize the clinical presentation and treat-ments attempted in these cases, and recommendations for future management. Please ameliorate this part of sentence "this review, we review..." for example "this reviewe, we evaluate ..." 

4) We further summarize the clinical presentation and treatments attempted in these cases, and recommendations for future management.  Please, improve this paragraph

5) Systemic Sclerosis/Scleroderma: L58-64 The term Scleroderma derives from the Greek roots skleros meaning hard or thick-  ened and derma indicating skin. When organ systems beyond the skin are impacted, the disease is categorized as systemic sclerosis (SSc). SSc is characterized by autoimmune-  mediated fibrosis of the skin and internal organs with a variable clinical presentation and  severity.5 SSc remains incompletely understood with a complex pathophysiology.5 There  are three main clinical subsets: (1) diffuse SSc (widespread sclerosis of the skin extending  proximal to the elbow/knee), (2) limited SSc (sclerosis is confined to the distal extremities)  and (3) SSc sine scleroderma (organ fibrosis without any skin involvement at all).5 While 66 the classification is based on the skin findings, the overall clinical course and prognosis also diverge between these groups. Diffuse SSc tends to have more severe progression of  skin fibrosis and higher incidence of interstitial lung disease. In order to discuss the previously described points, important references are needed to be added, such as: 

a- The Role of Bronchoalveolar Lavage in Systemic Sclerosis Interstitial Lung Disease: A Systematic Literature Review. Pharmaceuticals 2022, 15, 1584. https://doi.org/10.3390/ph15121584

b- The role of chest CT in deciphering interstitial lung involvement: systemic sclerosis versus COVID-19. Rheumatology (Oxford). 2022;61(4):1600-1609. doi:10.1093/rheumatology/keab615

6) Conclusion: The breadth of irAEs mediated disease states continues to expand as ICI use becomes more customary throughout the tumor spectrum. SSc and scleroderma mimics represent a group of disorders with significant morbidity that have been described with wide clinical presentation because of ICI use, with ICI use also being described as flaring of pSSc. Please, improve this paragraph.

7) Multicenter international clinical trials for treatment of these complications would be ideal to optimize treatment strategies and help forge this consensus. As use of these agents broadens and expands, this represents an exciting avenue of future research for the field of ICI induced autoimmunity. Please, improve this paragraph and underline the novelty of the study

Author Response

Thank you so much for taking the time to read our manuscript and for your valuable feedback. We sincerely appreciate the opportunity to re-submit our manuscript (2170743), entitled “Checkpoint inhibitor associated scleroderma and scleroderma mimics for re-consideration for publication in Pharmaceuticals.

Please see our responses under your reviewer suggestions and feedback below:

REVIEWER COMMENT 1:
There is a critical need for multi-institutional efforts to collaborate on developing a patient database for and conducting robust, prospective research on ICI-scleroderma. This will ultimately facilitate more effective clinical evaluation and management for ICI-scleroderma. Please underline the results of the review.

Authors:

Thank you for your comments. This was a segment taken from our Abstract. For that reason we understood that you wanted the results of our findings underlined in our Abstract. We have herewith cut-and-pasted the results of our review and underlined the key results:

“Cases of ICI-SSc had notable differences from primary SSc, such as fewer vascular features and less seropositivity (such as scleroderma-specific antibodies, antinuclear antibody).  We found that patients with a diagnosis of SSc prior to start of ICI can also experience flares of pre-existing disease after ICI treatment used for their cancer. Regarding scleroderma mimics, several cases of ICI-eosinophilic fasciitis have also been described with variable clinical presentation and course.”  

REVIEWER COMMENT 2:
L52-56. SSc and scleroderma  mimics are already rare entities, rendered less common still when exclusively selecting cases related to ICI use. Please, add a brief paragraph on systemic sclerosis (SSc) and scleroderma  mimics and insert this references:

    1. Correlation between circulating fibrocytes and dermal thickness in limited cutaneous systemic sclerosis patients: a pilot study. Rheumatol Int. 2019;39(8):1369-1376. doi:10.1007/s00296-019-04315-7
    2. Hyperpigmented Scleroderma-like Lesions under Combined Pembrolizumab and Pemetrexed Treatment of Non-Small Lung Cancer. Dermato20222, 8-13. https://doi.org/10.3390/dermato2010002

Authors:

References have been incorporated as suggested. Thank you for these suggestions. We did not add a separate paragraph in the introduction as under each section we do spend time giving an overview of each condition (scleroderma, morphea, eosinophilic fasciitis, etc.) to help orient the reader. References have been incorporated into these explanatory sections as relevant, mainly into the Systemic Sclerosis/Scleroderma section. Please see below sections:

“ILD and PAH represent the two major contributions of mortality from SSc, being responsible for 33% and 28% of deaths in SSc patients respectively.6  Classic physical examination findings include sclerodactyly (thickening of the fingers/toes) and the prayer sign (inability to fully extend the fingers when placing the palms together). Examples of sclerodactyly and the prayer sign in a patient with ICI related scleroderma in our center are shown in Figure 1.

Notably SSc can lead to significant esophageal dysmotility from fibrotic infiltration, with severe gastroesophageal reflux disease (GERD) symptoms that carry significant morbidity. There is emerging data that GERD influences the development of pulmonary fibrosis through damage from chronic aspiration of gastric contents with this likely representing at least part of the pathophysiology for SSc associated ILD.7 Key CT findings that can help discriminate SSc related ILD from other causes are fibrosis within ground glass opacities in the upper or lower lobes and reticulations within the lower lobes.8 SSc associated ILD is an active area of research with unmet need for additional therapeutic agents. Recently the anti-IL-6 monoclonal antibody tocilizumab was approved for slowing progression of SSc associated ILD based on the results of the focuSSed trial.9 The use of rituximab for SSc associated ILD also continues to evolve with a recent data from the RECITAL trial showing non inferiority to more traditionally used cyclophosphamide with less side effects.10 Notably tocilizumab has also been used for other irAEs including inflammatory arthritis, generally in cases refractory to tumor necrosis factor inhibitors.4,11

The exact etiology and pathways leading to SSc are still an area of rich and active research and not well understood. Fibroblast dysregulation with increased extracellular matrix (ECM) secretion resulting in fibrosis of tissues/organs is a key feature of SSc and is thought to be due to immune related mechanisms.12 Fibroblasts being a key cell type involved in skin thickening in SSc is supported by a correlation of higher circulating fibroblast levels in blood samples from patients with higher modified Rodan skin scores (mRSS) and dermal thickness when measured by ultrasound.12

Outside of ICI therapy for malignancy, several other chemotherapeutic agents used for various malignancies have been reported to lead to scleroderma or morphea like skin reactions. This includes the taxane drug class, pemetrexed, gemcitabine, and bleomycin.13 The mechanisms for these reactions are poorly understood though vesican; a large ECM proteoglycan molecule that serves an immune cell trafficking and activation role appears to be involved in taxane induced scleroderma/morphea skin involvement.14,15

REVIEWER COMMENT 3:
In this review, we review the specific role of immune checkpoints in the  pathogenesis of scleroderma. We further summarize the clinical presentation and treatments attempted in these cases, and recommendations for future management. Please ameliorate this part of sentence "this review, we review..." for example "this review, we evaluate ..." 

Authors:

The wording has been changed. Please see below.

“In this review, we evaluate how immune checkpoints are involved in the pathogenesis of scleroderma by going through the molecular pathways of immune checkpoints. We also present an overview of scleroderma, and its mimics and survey the available literature for ICI-related cases by  summarizing the clinical presentation and treatments attempted in these cases. Finally, when possible we also suggest recommendations for future management of these ICI-related cases.”

REVIEWER COMMENT 4:
We further summarize the clinical presentation and treatments attempted in these cases, and recommendations for future management.  Please, improve this paragraph

Authors:

The wording has been changed. Please see below.

“In this review, we evaluate how immune checkpoints are involved in the pathogenesis of scleroderma by going through the molecular pathways of immune checkpoints. We also present an overview of scleroderma, and its mimics and survey the available literature for ICI induced/related cases by  summarizing the clinical presentation and treatments attempted in these cases. Finally, when possible we also suggest recommendations for future management of these ICI induced/related cases.”

REVIEWER COMMENT 5:
Systemic Sclerosis/Scleroderma: L58-64 The term Scleroderma derives from the Greek roots skleros meaning hard or thick-  ened and derma indicating skin. When organ systems beyond the skin are impacted, the disease is categorized as systemic sclerosis (SSc). SSc is characterized by autoimmune-  mediated fibrosis of the skin and internal organs with a variable clinical presentation and  severity. SSc remains incompletely understood with a complex pathophysiology. There  are three main clinical subsets: (1) diffuse SSc (widespread sclerosis of the skin extending  proximal to the elbow/knee), (2) limited SSc (sclerosis is confined to the distal extremities)  and (3) SSc sine scleroderma (organ fibrosis without any skin involvement at all).While the classification is based on the skin findings, the overall clinical course and prognosis also diverge between these groups. Diffuse SSc tends to have more severe progression of  skin fibrosis and higher incidence of interstitial lung disease. In order to discuss the previously described points, important references are needed to be added, such as: 

    1. The Role of Bronchoalveolar Lavage in Systemic Sclerosis Interstitial Lung Disease: A Systematic Literature Review. Pharmaceuticals 2022, 15, 1584. https://doi.org/10.3390/ph15121584
    2. The role of chest CT in deciphering interstitial lung involvement: systemic sclerosis versus COVID-19. Rheumatology (Oxford). 2022;61(4):1600-1609. doi:10.1093/rheumatology/keab615

Authors:

Suggested references have been added. In general background info on SSc has been significantly expanded per suggestions with additional references added as well. Please see below. Thank you for the suggested references.

“For both diffuse and limited SSc, classic physical examination findings include sclerodactyly (thickening of the fingers/toes) and the prayer sign (inability to fully extend the fingers when placing the palms together) as demonstrated in Figure 1. Aside from sclerodactyly, SSc can lead to significant esophageal dysmotility from fibrotic infiltration, with severe gastroesophageal reflux disease (GERD) symptoms that carry significant morbidity. There is emerging data that GERD influences the development of pulmonary fibrosis through damage from chronic aspiration of gastric contents with this likely representing at least part of the pathophysiology for SSc associated ILD.7 Key CT findings that can help discriminate SSc-related ILD from other causes are fibrosis within ground glass opacities in the upper or lower lobes and reticulations within the lower lobes.8

Overall, the exact etiology and pathways leading to SSc are still an area of rich and active research and not well understood. Fibroblast dysregulation with increased extracellular matrix (ECM) secretion resulting in fibrosis of tissues/organs is a key feature of SSc and is thought to be due to immune related mechanisms.11 Fibroblasts being a key cell type involved in skin thickening in SSc is supported by a correlation of higher circulating fibroblast levels in blood samples from patients with higher modified Rodan skin scores (mRSS) and dermal thickness when measured by ultrasound.11

While we focused our review on that of ICI-associated scleroderma, it should be noted that several other oncologic agents such as various chemotherapies have also been associated with scleroderma or morphea-like skin reactions. This includes the taxane drug class, pemetrexed, gemcitabine, and bleomycin.12 The mechanisms for these reactions are poorly understood though vesican; a large ECM proteoglycan molecule that serves an immune cell trafficking and activation role appears to be involved in taxane induced scleroderma/morphea skin involvement.13,14

For the purposes of this review that focuses on ICI-SSc and its mimics, SSc that is diagnosed prior to start of ICI therapy will be known as primary systemic sclerosis or pre-existing systemic sclerosis and abbreviated as pSSc. On the other hand, SSc that is diagnosed after start of ICI therapy will be known as immune checkpoint-associated SSc or ICI-SSc. In this review, we will also discuss cases of ICI-associated morphea as well as scleroderma mimics such as eosinophilic fasciitis, morphea, scleromyxedema and scleroedema.”

REVIEWER COMMENT 6:
Conclusion: The breadth of irAEs mediated disease states continues to expand as ICI use becomes more customary throughout the tumor spectrum. SSc and scleroderma mimics represent a group of disorders with significant morbidity that have been described with wide clini[1]cal presentation because of ICI use, with ICI use also being described as flaring of pSSc. Please, improve this paragraph.

Authors:

Some modifications have been made to the conclusion. Please see below.

“Scleroderma mimics such as morphea and eosinophilic fasciitis have been reported with a similarly wide range of presentations, treatments and clinical outcomes., No ICI related cases of but scleromyxedema and scleroedema were found in our literature review. However, due to their association with malignancy and other common comorbidities, these entities should be included on the differential in a patient that develops skin thickening during ICI use to avoid unneeded and potentially harmful changes to their cancer treatment regimens.”

REVIEWER COMMENT 7:
Multicenter international clinical trials for treatment of these complications would be ideal to optimize treatment strategies and help forge this consensus. As use of these agents broadens and expands, this represents an exciting avenue of future research for the field of ICI induced autoimmunity. Please, improve this paragraph and underline the novelty of the study

Authors:

Thank you for this suggestion. In response, modifications have been made to our conclusion. In a nutshell, our goal/intent is to have a comprehensive review on the subject providing a useful reference for future research and currently practicing physicians. Currently in the literature no large reviews exist on this topic with case series/reports with incomplete literature reviews. Our review represents a comprehensive summary of useful information for these purposes. Please see updated conclusion cut-and-pasted here:

“Conclusion:

The breadth of irAEs mediated disease states continues to expand as ICI use becomes more customary throughout the tumor spectrum. SSc and scleroderma mimics represent a group of disorders with significant morbidity that have been described with wide clinical presentation because of ICI use, with ICI use also being described resulting in  flaring of pSSc. Oncologists, dermatologists, and rheumatologists should be aware of these potential ICI toxicities. Compared with pSSc, ICI-SSc induced cases appear to have fewer vascular features with reduced frequency of Raynaud’s, as well as lower levels of scleroderma specific antibody and ANA positivity. Cases to date suggest PD-1/PD-L1 blockade to be higher risk than CTLA-4 inhibition. ICI use has also have been described to cause flares of pSSc. Scleroderma mimics such as morphea and eosinophilic fasciitis have been reported with a similarly wide range of presentations, treatments and clinical outcomes. No ICI related cases of  scleromyxedema and scleroedema were found in our literature review. However, due to their association with malignancy and other common comorbidities, these entities should be included on the differential in a patient that develops skin thickening during ICI use to avoid unneeded and potentially harmful changes to their cancer treatment regimens.

Further study and reporting of cases will help us understand whether the clinical finding that most ICI-SSc has a lower burden of vascular features and autoantibody positivity compared with primary SSc cases remains consistent. Our review serves as a starting point on this topic and represents a complete and comprehensive summary of the field’s current knowledge and description of ICI related SSc and scleroderma mimics. Our hope is that this review will be a useful reference for currently practicing physicians who treat ICI related irAEs, representing a practical source of information compared with perusing the of dozens of individual case reports/series in the literature. Such a source would also be useful in spurring further research on this topic. Treatment guidance at this time is limited, ranging from continuation of ICI to drug cessation and addition of immunosuppression. We recommend extrapolating from the standardized management of pSSc and related conditions, with the decision to hold or resume ICI decided on a case-by-case basis weighing the severity of the reaction against the options and outcomes expected for the underlying cancer. As more cases are described we would look forward to consensus guidelines on management of ICI induced scleroderma, morphea and eosinophilic fasciitis. Multicenter international clinical trials for treatment of these complications would be ideal to optimize treatment strategies and help forge this consensus. As use of these agents broadens and expands, this represents an exciting avenue of future research for the field of ICI induced autoimmunity.”

Round 2

Reviewer 1 Report

My opinion about the article remains the same. Most of the issues that I advanced to you not  repaired. A new study would be needed to make these things suitable. The clarifications provided do not solve the problem. Best regards.